# Fabrication of High-Quality MoS_2_/Graphene Lateral Heterostructure Memristors

**DOI:** 10.3390/nano15161239

**Published:** 2025-08-13

**Authors:** Claudia Mihai, Iosif-Daniel Simandan, Florinel Sava, Teddy Tite, Amelia Bocirnea, Mirela Vaduva, Mohamed Yassine Zaki, Mihaela Baibarac, Alin Velea

**Affiliations:** National Institute of Materials Physics, Atomistilor 405A, 077125 Magurele, Romania; claudia.mihai@infim.ro (C.M.); simandan@infim.ro (I.-D.S.); fsava@infim.ro (F.S.); teddy.tite@infim.ro (T.T.); amelia.bocirnea@infim.ro (A.B.); mirela.ilie@infim.ro (M.V.); yassine.zaki@infim.ro (M.Y.Z.); barac@infim.ro (M.B.)

**Keywords:** MoS_2_/graphene heterostructure, confined sulfurization, lateral memristor, few-layer MoS_2_, vacancy-assisted switching, sputter deposition, neuromorphic electronics

## Abstract

Integrating two-dimensional transition-metal dichalcogenides with graphene is attractive for low-power memory and neuromorphic hardware, yet sequential wet transfer leaves polymer residues and high contact resistance. We demonstrate a complementary metal–oxide–semiconductor (CMOS)-compatible, transfer-free route in which an atomically thin amorphous MoS_2_ precursor is RF-sputtered directly onto chemical vapor-deposited few-layer graphene and crystallized by confined-space sulfurization at 800 °C. Grazing-incidence X-ray reflectivity, Raman spectroscopy, and X-ray photoelectron spectroscopy confirm the formation of residue-free, three-to-four-layer 2H-MoS_2_ (roughness: 0.8–0.9 nm) over 1.5 cm × 2 cm coupons. Lateral MoS_2_/graphene devices exhibit reproducible non-volatile resistive switching with a set transition (SET) near +6 V and an analogue ON/OFF ≈2.1, attributable to vacancy-induced Schottky-barrier modulation. The single-furnace magnetron sputtering + sulfurization sequence avoids toxic H_2_S, polymer transfer steps, and high-resistance contacts, offering a cost-effective pathway toward wafer-scale 2D memristors compatible with back-end CMOS temperatures.

## 1. Introduction

The extraordinary electronic, optical, and mechanical properties of two-dimensional (2D) van der Waals materials have stimulated intense research into heterostructures that integrate complementary layers within a few atomic planes. Graphene provides ultrahigh room-temperature carrier mobility (10^3^–10^4^ cm^2^ V^−1^ s^−1^) but lacks an intrinsic band-gap, whereas single-layer MoS_2_ displays a direct gap of ≈1.8 eV, together with strong light–matter interaction and sizable spin–valley coupling [1]. Stacking MoS_2_ on graphene therefore combines a gate-tunable semiconducting channel with a transparent, low-resistance electrode. For instance, inserting a graphene interlayer between Ag contacts and chemical-vapor deposition (CVD) MoS_2_ raised the field-effect mobility to 35 cm^2^ V^−1^ s^−1^ and boosted photoresponsivity to 2160 A W^−1^, four orders of magnitude above devices with conventional Ti/Au contacts [2]. Beyond charge-injection benefits, interfacial coupling can modulate the quasiparticle gap of MoS_2_ from ≈2.23 eV to 1.98 eV, providing a route to band-gap engineering unavailable in the isolated constituents [3].

At the same time, both layers are mechanically robust, bendable, and lattice mismatch-tolerant, making MoS_2_/graphene stacks attractive for flexible opto-electronics and memory applications [4]. Recent first-principle modelling further highlights the versatility of such van der Waals stacks, showing that Ni-decorated WS_2_/WSe_2_ heterostructures can serve as highly sensitive, reusable gas sensors for C_2_H_2_ and C_2_H_4_ in transformer oil diagnostics [5]. Because the Fermi level of graphene can be electrostatically or chemically tuned, MoS_2_/graphene junctions offer a platform for adjustable Schottky barriers that are attractive for photodetection, neuromorphic weighting, and low-power logic [6].

Translating these attributes into wafer-scale devices requires fabrication strategies that deliver large-area, crystalline MoS_2_/graphene interfaces with minimal contamination. Layer-by-layer CVD can, in principle, realize such heterostructures; Kim et al. reported monolayer MoS_2_ grown on graphene-coated sapphire across a four-inch wafer via metal–organic CVD [7]. However, multi-step CVD demands costly reactors and stringent integration sequences to avoid degrading the first-grown layer. Sequential wet transfer (e.g., polymethyl methacrylate (PMMA)-assisted lift-off of CVD graphene, followed by stacking on CVD MoS_2_), is simpler but introduces cracks, wrinkles, and polymer residues that pin the Fermi level and degrade mobility [8]. Polymer-free dry-transfer methods using visco-elastic or h-BN stamps achieve atomically clean interfaces and record mobilities [9], but they remain restricted to microscope-scale flakes.

Physical-vapor routes that decouple film formation from crystallization have emerged as a pragmatic alternative. In a “sputter-then-sulfurize” scheme, a metal Mo or amorphous MoS_2_ precursor is deposited at room temperature and subsequently annealed in a sulfur-rich atmosphere to yield crystalline MoS_2_. Li et al. converted ultrathin Mo films to monolayer MoS_2_ by enclosing the sample and sulfur powder in a sealed quartz sandwich, obtaining centimeter-scale single crystals [10]. Graphite-box confinement further improves sulfur utilization and suppresses oxidation, but most demonstrations have focused on MoS_2_ films grown on inert substrates; the direct growth of high-quality MoS_2_ on graphene remains scarcely explored.

In parallel, MoS_2_-based memristors have attracted attention for non-volatile memory and neuromorphic computing. Despite rapid progress, MoS_2_-based memristors continue to face critical material-integration challenges. Vertical metal/MoS_2_/metal (MIM) stacks can deliver ON/OFF ratios exceeding 10^4^ but require an initial electroforming step and often degrade by stochastic filament growth at high current densities [11]. Lateral planar architectures dispense with forming and allow in-plane neuromorphic weighting, yet most rely on polymer-assisted transfer of MoS_2_ or graphene, leaving behind residues that pin the Fermi level, raise contact resistance, and limit endurance [12]. Direct CVD growth on graphene has produced wafer-scale monolayers, but temperatures above 750 °C exceed the 450 °C back-end-of-line (BEOL) ceiling of advanced complementary metal–oxide–semiconductors (CMOSs) [13] and risk sulfur diffusion into Cu interconnects [14]. Ionic-liquid gating can lower operating voltages at the expense of process complexity and volatility [15]. Consequently, a transfer-free, BEOL-compatible route capable of Å-level thickness control and polymer-free MoS_2_/graphene interfaces remains elusive. Lateral two-terminal devices based on monolayer MoS_2_ reported by Shen et al. switch via sulfur-vacancy drift with ON/OFF ratios below ten and set voltages of ~6 V [16]. Xiong et al. achieved similar barrier-modulated behavior in oxygen-doped MoS_2_ with ±5 V sweeps [17]. Graphene electrodes can stabilize these trap-assisted mechanisms while providing low series resistance, as shown by Dragoman et al. in MoS_2_ self-switching diodes [18]. A scalable method that forms the MoS_2_ channel directly on graphene would therefore streamline fabrication of memristive cross-bars.

Here, we report a cost-effective two-step route, Å-level RF sputtering of an amorphous MoS_2_ precursor, followed by confined-space sulfurization at 800 °C, that generates centimeter-scale, few-layer MoS_2_ directly on chemical vapor-deposited graphene. The confined-sulfurization strategy reported here addresses the existing gaps by converting an RF-sputtered amorphous precursor into a uniform three-to-four-layer 2H-MoS_2_ sheet directly on graphene at 800 °C, well below front-end peak temperatures. The process precludes polymer contamination, preserves graphene conductivity, and yields forming-free, interface-controlled switching suitable for analog neuromorphic hardware. Comprehensive structural and chemical characterization confirms the formation of uniform, three-to-four-layer 2H-MoS_2_ with near-stoichiometric composition and sub-nanometer roughness. Lateral Au/MoS_2_/graphene devices exhibit reproducible, non-volatile resistive switching with a threshold of ~6 V and ON/OFF ≈ 2.1, consistent with vacancy-assisted Schottky-barrier modulation rather than filament formation.

## 2. Materials and Methods

### 2.1. Substrate Preparation and Graphene Growth

Commercial 100 mm Si wafers with 300 nm thermally grown SiO_2_ (MicroChemicals GmbH, Ulm, Germany) were diced into 15 mm × 20 mm coupons. Coupons were sequentially rinsed in acetone, isopropanol, and de-ionized (DI) water (5 min each in an ultrasonic bath), and then they were finally exposed for 2 min to a low-power (50 W, 200 mTorr) O_2_ plasma to remove organic residues.

Few-layer graphene (FLG) was synthesized on 25 µm Cu foil (99.8%, Alfa Aesar, Massachusetts, U.S.) in an AS-ONE rapid thermal CVD reactor (Annealsys, Montpellier, France). After H_2_ annealing (1000 °C, 50 sccm, 10 mbar, 10 min), the foil was exposed to CH_4_:H_2_ = 24:8 sccm at 900 °C for 10 min, leading to three to four graphene layers [19].

Graphene was transferred to the SiO_2_/Si coupons using a polymer-assisted bubbling method [19]. A 400 nm PMMA (950 K, A4) layer was spin-cast (4000 rpm, 60 s) and soft-baked (65 °C, 3 min). Electrochemical delamination was carried out in 0.5 M NaCl with the PMMA/graphene/Cu stack as cathode (–1.8 V vs. Ag/AgCl) and a Pt mesh as anode. The floating PMMA/graphene film was rinsed in DI water (3×) and placed onto the target SiO_2_/Si substrate, dried at 65 °C (10 min) and 150 °C (30 min), and the PMMA was removed in hot acetone (50 °C, 2 h). Raman mapping of transferred films yielded an I_2D_/I_G_ ratio of ≈0.85, consistent with three to four graphene layers [4].

### 2.2. Amorphous MoS_2_ Precursor Deposition

Ultrathin amorphous MoS_2_ (a-MoS_2_) precursor films were deposited in a custom-built cylindrical sputtering chamber [20]. A 50 mm MoS_2_ target (99.9%, Mateck, Jülich, Germany) was ignited in RF mode at 16 W, while 30 sccm Ar maintained 5 × 10^−3^ Torr. The FLG/SiO_2_/Si coupon was mounted 11 cm above the target on a rotating holder (20 rpm) to ensure uniformity. Step-height measurements by AFM yielded a growth rate of 0.027 nm s^−1^; deposition times of 30 s and 37 s therefore produced 0.8 nm and 1.0 nm a-MoS_2_, respectively. For heterostructure samples, an aluminum shadow mask shielded one-third of the graphene, yielding discrete graphene-only overlap (MoS_2_/graphene) and MoS_2_-only zones (Figure 1d).

### 2.3. Confined-Space Sulfurization

Sulfurization was conducted in an MTI GSL-1600X three-zone furnace equipped with a 50 mm ID fused-quartz tube. Each sample, together with 25 mg elemental sulfur (99.5%, Sigma-Aldrich, Darmstadt, Germany), was sealed in a graphite box to maintain a high local sulfur vapor pressure [21]. After three pump-and-purge cycles (base pressure: 5 × 10^−2^ mbar), the tube was back-filled with 100 sccm Ar to 0.02 MPa and heated to 800 °C at 10 °C min^−1^. A 45 min dwell yielded complete conversion of a-MoS_2_ to crystalline 2H-MoS_2_ while preserving the underlying graphene. The 800 °C dwell was selected because it lies above the amorphous-to-2H crystallization threshold yet below the 1T metastability window. Absence of alkali intercalants further suppresses the metallic 1T phase, as verified by the E^1^_2g_/A_1g_ Raman doublet and the Mo 3d binding energy of 228.5 eV characteristic of 2H-MoS_2_. The sealed graphite box serves two purposes: (i) it traps sublimed sulfur in the immediate vicinity of the sample, sustaining a high local sulfur chemical potential even as the outer quartz tube is purged; and (ii) in combination with the Ar back-fill to 0.02 MPa (~150 Torr), it establishes a moderate total pressure regime that mitigates sulfur-vacancy formation during the 800 °C dwell. Islam et al. systematically varied sulfurization pressure (50–300 Torr) for Mo thin films and showed that higher pressures suppress vacancy-related Raman satellites, narrow the A_1g_ linewidth, and improve electronic transport [22]. Related confined-space or sealed sulfurization approaches have been reported to enhance crystallinity and stoichiometry in magnetron sputtering-derived or vapor-converted ultrathin MoS_2_ films [21,23,24]. This approach maximizes sulfur availability, preserves the graphene underlayer, and avoids oxide formation.

### 2.4. Structural and Chemical Characterization

X-ray reflectivity (XRR), grazing-incidence XRD (Vertical GIXRD, θ_i_ = 0.2°), and grazing-incidence in-plane XRD (Horizontal GIXRD, θ_i_ = θ_d_ = 0.2°) were recorded on a Rigaku SmartLab diffractometer equipped with a Cu K_α_ source (λ = 1.54178 Å) and a HyPix-3000 0D detector (Rigaku Corporation, Tokyo, Japan).

Raman spectra and maps were obtained with a LabRAM HR Evolution spectrometer (Horiba Jobin Yvon, Palaiseau, France), using 532 nm and 633 nm excitations, through an Olympus 100× objective (NA = 0.9). Laser power was kept below 1 mW; the Si phonon at 519.8 cm^−1^ provided the calibration. Spectra were fitted with Voigt profiles to extract the E^1^_2g_ and A_1g_ positions for layer counting [1].

Topography was acquired in non-contact mode (NT-MDT Aura) with 10 µm × 10 µm scans; areal roughness S_q_ was calculated with Gwyddion v2.69. Surface morphology and composition were examined on a ZEISS Gemini-500 FE-SEM (3 kV, InLens detector, Zeiss, Oberkochen, Germany) equipped with a Bruker EDS system (Billerica, Massachusetts, U.S.).

Surface chemical states were examined with a Kratos AXIS Ultra DLD XPS system (Kratos Analytical Ltd., Manchester, UK, monochromatic Al K_α_, 12 kV × 12 mA, pass energy 20 eV). Spectra were charge-neutralized in situ and aligned to the graphene C 1s sp^2^ signal at 284.6 eV; Voigt profiles and Shirley backgrounds were fitted in CasaXPS v2.3.

### 2.5. Device Fabrication and Electrical Testing

Metal contacts were defined by shadow-mask sputtering: 5 nm Ti/30 nm Au was deposited through precision stainless-steel masks (Ossila, Sheffield, UK). Current–voltage characteristics were measured under ambient conditions with a Keithley 4200A-SCS (Cleveland, Ohio, US); low-bias sweeps employed ±3 V at 0.1 V s^−1^, whereas resistive-switching measurements used ±7 V with a 10 mA compliance limit.

## 3. Results

### 3.1. Process Scheme and Heterostructure Architecture

Figure 1 summarizes the two-step route adopted to obtain high-quality, centimeter-scale MoS_2_/graphene heterostructures on SiO_2_/Si. In the first step (Figure 1a), an ultrathin amorphous MoS_2_ (a-MoS_2_) precursor is radio frequency-sputtered onto chemical vapor-deposited few-layer graphene (FLG, ≈1.3 nm, three to four layers). A confined-space sulfurization at 800 °C (Figure 1b,c) then crystallizes the precursor into 2H-MoS_2_ while fully preserving the underlying graphene, a strategy that minimizes polymer or oxide contamination compared with conventional wet-transfer stacking [21].

The resulting coupon is deliberately partitioned into three lateral zones, graphene-only, heterostructure (MoS_2_/graphene), and MoS_2_-only, using a shadow mask during precursor deposition (Figure 1d). This geometry enables direct, same-wafer comparison of (i) the intrinsic graphene sheet resistance, (ii) the electrical behavior of the heterointerface, and (iii) the contribution of the stand-alone MoS_2_ film. Such a design has been advocated as the best practice for 2D heterostructure electronics [25].

Compared with sequential wet-transfer assembly, the present route offers three advantages. First, the magnetron sputtering + sulfurization sequence is compatible with large-area industrial tooling, avoids the handling of toxic H_2_S, and yields highly uniform films over at least 1.5 cm × 2 cm. Second, the van der Waals interface between MoS_2_ and graphene is formed in situ, obviating polymer residues that notoriously pin the Fermi level in transferred stacks [26]. Third, graphene acts as an atomically flat growth template and as a built-in current spreading layer, both of which are beneficial for the electrical robustness of laterally configured memristors.

### 3.2. Thickness Calibration and Crystallization of Ultrathin MoS_2_

Figure 2 corroborates the film thickness obtained from the sputter-time calibration and follows the structural evolution of the amorphous precursor during confined sulfurization. Atomic force-microscopy step-height measurements (Figure 2a) yield a linear growth rate of 0.027 nm s^−1^; sputtering for 30 s and 37 s therefore deposits ≈ 0.8 nm and ≈1.0 nm of amorphous MoS_2_, respectively.

The corresponding X-ray-reflectivity profiles (Figure 2b) display only faint, rapidly damped Kiessig fringes in the as-deposited state, indicative of a low-density, highly disordered film. After sulfurization, the fringe amplitude increases, and the oscillation period shifts slightly toward higher scattering vectors, signifying densification of the MoS_2_ layer and a modest contraction of the overall film thickness as the amorphous network relaxes into the more closely packed 2H lattice. Because the residual thickness is close to the critical value at which XRR fitting becomes unreliable for Cu K_α_ radiation, no numerical thickness or density values are extracted; nonetheless, the qualitative changes are entirely consistent with a transition from an amorphous to a crystalline, continuous sheet.

We note that sulfur availability and ambient pressure strongly influence the degree of crystallization and the density of sulfur vacancies in MoS_2_ formed from sputtered precursors. Islam et al. reported that raising the sulfurization chamber pressure from 50 to 300 Torr markedly reduced defect-related Raman satellites and produced smoother, more conductive MoS_2_ films [22]. Our confined graphite-box geometry, operated under an Ar back-pressure of 0.02 MPa (~150 Torr), is intended to mimic the “high-pressure/high-S” regime that favors near-stoichiometric conversion. This rationale is consistent with prior confined or high-S conversions of sputter-derived Mo or MoS_2_ precursors that achieved improved 2H texture and device performance [21,23,24]. The stoichiometric Mo:S ratios and strong Raman response we observe after sulfurization indicate that the approach is effective even for sub-nanometer starting thicknesses.

The crystalline signal obtained from grazing-incidence X-ray diffraction is strongly constrained by the extreme thinness of the present films. In the vertical configuration (incidence angle 0.20°), both the 0.8 nm and 1.0 nm sulfurized samples exhibit a broad, low-intensity background that is dominated by the amorphous SiO_2_ layer and the Si substrate (Figure 2c, top). Within the signal-to-noise limits of the measurement, no unambiguous (002) reflection of 2H-MoS_2_ can be distinguished at 2θ ≈ 13° for either film. The absence of clear peaks in this geometry therefore reflects the detection limit of the technique for ultrathin (<2 nm) layers rather than proof that the material remains amorphous.

In the horizontal (in-plane) geometry, where the detector is scanned in the sample surface plane, weak diffraction maxima attributable to the (100) and (110) families of 2H-MoS_2_ are discernible for the 1.0 nm specimen, whereas they remain below the noise level for the 0.8 nm film (Figure 2d). The appearance of these in-plane reflections, albeit faint, implies that the 1.0 nm layer has crystallized with the van der Waals planes parallel to the graphene substrate, consistent with the ⟨00l⟩ texture reported for similarly thin sputter-derived MoS_2_ films after sulfurization [27].

### 3.3. Raman Confirmation of Crystallinity and Spatially Selective Heterostructure Formation

Raman spectra provide a highly sensitive probe to both layer number and structural order for ultra-thin MoS_2_ films that are close to the X-ray detection limit and for evaluating any damage to the underlying graphene. Figure 3a compares 532 nm spectra collected on the 0.8 nm and 1.0 nm specimens before and after sulfurization. The amorphous precursors show only the Si substrate mode at 519 cm^−1^, whereas the annealed films reveal the two first-order modes of 2H-MoS_2_, E^1^_2g_ at 385 ± 1 cm^−1^ and A_1g_ at 406 ± 1 cm^−1^. The mode separation, Δω ≈ 21 cm^−1^, corresponds to three to four S–Mo–S trilayers, in agreement with calibrated AFM step heights [1].

Excitation at 633 nm (Figure 3b) enhances multiphonon processes. Two resonant acoustic combinations are observed at 178 cm^−1^ [LA(M) −TA(M)] and 227 cm^−1^ [first-order LA(M)], followed by the weak p_1_ band at 416 cm^−1^ [LA(K) + TA(K)] and a dominant 2LA(M) overtone at 456 ± 2 cm^−1^. These second-order features require long-range crystalline order and confirm that even the 0.8 nm film is fully crystallized, although its X-ray signature is below the GIXRD detection limit [28,29]. A separate α (A_1g_ + E^2^_2g_) band at ≈440 cm^−1^ is not resolved, a common situation for few-layer MoS_2_ measured with moderate laser power.

Figure 3c displays 532 nm spectra acquired at three lateral positions on the 0.8 nm coupon: graphene-only, overlap (MoS_2_/graphene), and MoS_2_-only. The graphene zone shows the expected G (1582 cm^−1^) and 2D (2700 cm^−1^) bands together with a moderate D band (≈1350 cm^−1^). The intensity ratio I_D_/I_G_ ≈ 0.90 yields an in-plane crystallite size of L_a_ ≈ 22 nm according to the Tuinstra–Koenig relation recalibrated for 532 nm excitation [30]. The essentially identical ratio measured in the MoS_2_/graphene overlap indicates that the sputter–sulfurization sequence does not introduce additional, local disorder into graphene beyond that already set by the high-temperature exposure and the initial CVD transfer. A shoulder at ≈1446 cm^−1^, pronounced in the graphene spectrum, weak in the overlap, and absent in the MoS_2_-only area, originates from the 2TO phonon of the Si substrate and is attenuated as the overlayer becomes optically opaquer.

Identical trends are observed on the 1.0 nm coupon (Figure 3d). The spatially selective coexistence of graphene and MoS_2_ modes confirms that the shadow-mask strategy yields sharp lateral heterojunctions, while the constant Δω across the coupon demonstrates thickness uniformity, in agreement with the AFM maps (Figure 4a,b).

### 3.4. Surface Morphology and Stoichiometry of the Sulfurized MoS_2_ Films and the MoS_2_/Graphene Heterostructure

Atomic force-microscopy height maps (Figure 4a,b) demonstrate that both the 0.8 nm and 1.0 nm sulfurized films form continuous, featureless blankets over the centimeter-scale coupon. The areal root mean-square roughness is S_q_ = 0.81 nm and S_q_ = 0.85 nm, respectively, values typical for few-layer MoS_2_ grown on atomically flat supports and fully compatible with the roughness window reported for sputter-converted MoS_2_ on sapphire (0.7–1.0 nm) [31]. No pin-holes or line defects are observed, confirming that the densification inferred from XRR extends laterally across the entire film.

A representative high-resolution SEM micrograph of the overlap region (Figure 4c) reveals a compact nanograined surface composed of 30–50 nm crystallites. Such morphology is characteristic of ultrathin sputter-derived MoS_2_ that recrystallizes under sulfur-rich conditions and has previously been correlated with reduced trap density in memristive devices [27]. Importantly, no cracks, delamination, or graphene wrinkles emerge after the 800 °C treatment, underscoring the mechanical compatibility of the two-step process.

The corresponding energy-dispersive X-ray spectrum, collected at 20 kV on the same area (Figure 4d), is dominated by the overlapping Mo L_α_ (2.293 keV) and S K_α_ (2.307 keV) lines. Quantification after overlap correction gives Mo:S atomic ratios of 1:1.2 for the 0.8 nm film and 1:1.5 for the 1.0 nm film. These sulfur-deficient values are typical for ultrathin MoS_2_ analyzed by EDS, where the interaction volume is small and the Mo/S peak overlap systematically underestimates sulfur. Importantly, no foreign elements are detected above the noise level, confirming that the confined-space sulfurization introduces no extraneous contamination.

### 3.5. Chemical-State Analysis by X-Ray Photoelectron Spectroscopy

High-resolution X-ray photoelectron spectroscopy (XPS) clarifies both the chemical purity of the confined-space process and the extent of sulfurization achieved in the ultrathin films. In the Mo 3d + S 2s window (Figure 5a), the 1 nm and the 0.8 nm samples are dominated by the Mo 3d_5/2_/3d_3/2_ doublet at 228.5/231.6 eV, characteristic of Mo^4+^ in 2H-MoS_2_ [28]. A weak high-binding-energy shoulder centered at 232.7 eV is assigned to surface MoO_3_ that forms upon brief air exposure; quantitative fitting yields Mo^6+^ fractions of 22% (1 nm) and 14% (0.8 nm). Importantly, no metallic Mo or sub-stoichiometric MoO_x_ signals are detected, confirming complete sulfur incorporation throughout the film thickness.

The S 2p spectra (Figure 5b) consist exclusively of the S^2−^2p_3/2_/2p_1/2_ doublet at 161.6/162.8 eV; no sulphate-related shoulders appear above 168 eV, corroborating the absence of adventitious oxidation during annealing. The Mo:S atomic ratios extracted from peak areas, 1:1.95 (1 nm) and 1:1.50 (0.8 nm), deviate slightly from the ideal 1:2, consistent with the scatter typically reported for mono-/few-layer MoS_2_ where XPS quantification is affected by overlayer attenuation and local non-stoichiometry [32]. Importantly, no Na, Cl, or Cu residues from the graphene transfer are observed above the detection limit (<0.2 at %), underscoring the chemical cleanliness of the overall fabrication sequence. See Table 1 for quantitative peak parameters.

The confined sulfur flux efficiently converts the amorphous precursor without introducing foreign contaminants; the modest surface oxide is typical for sputter-derived MoS_2_ films stored in ambient conditions and can be removed by a brief 200 °C Ar/H_2_ anneal.

### 3.6. Electrical Transport and Memristive Behavior

Figure 6 compares the low-bias conductance of the three lateral paths patterned on each coupon with the high-field switching response of the MoS_2_/graphene heterostructure. At low bias (±3 V, Figure 6a), the current–voltage characteristics are plotted on a semi-logarithmic scale to emphasize the wide dynamic range. The graphene–graphene trace (black, Figure 6a) is quasi-ohmic for forward bias with a sheet resistance of 1.9 kΩ sq^−1^. This falls in the 1–5 kΩ sq^−1^ range typical for transferred, few-layer CVD graphene with an in-plane crystallite size L_a_ ≈ 20 nm (as inferred from I_D_/I_G_ ≈ 0.9) [30].

The MoS_2_–MoS_2_ path (red) is two orders of magnitude less conductive with a channel resistance of ≈300 kΩ and a sheet resistance of ≈4 × 10^5^ Ω sq^−1^, in line with lateral transport through a 2 nm thick 2H-MoS_2_ sheet. The mixed graphene–MoS_2_ configuration (blue) lies between these limits and exhibits a small volatile hysteresis, the reverse sweep retraces a slightly lower current than the forward sweep, consistent with reversible Schottky-barrier modulation at the MoS_2_/graphene interface [25].

When the bias on the graphene–MoS_2_ pair is extended to ±7 V (Figure 6b), a SET transition occurs at ≈+6 V, with an ON/OFF ratio of ≈2.1. The non-volatile change is reproducible over several cycles and is characteristic of interfacial charge-trap mechanisms [33]. The reverse sweep to −7 V partially resets the device, consistent with vacancy detrapping or back-diffusion, as illustrated in Figure 7. The optical micrographs in the inset of Figure 6a,b identify the electrode pairs used; contact pads were defined by Ti/Au shadow-mask sputtering. All measurements were performed in ambient air, at room temperature, with a 10 mA compliance to prevent device breakdown.

### 3.7. Proposed Switching Mechanism

The electrical data of Figure 6 are consistent with a trap-assisted, interface-controlled mechanism rather than with metallic filament formation. Under a forward bias of +6 V, the electric field across the few-layer MoS_2_ is on the order of 3 MV cm^−1^, sufficient to drift pre-existing sulfur vacancies V_s_ toward the MoS_2_/graphene boundary, where they accumulate and locally pin negative charge (Figure 7, left → right). Density-functional calculations show that chains of sulfur vacancies separated by less than 0.5 nm create mid-gap states that pin the Fermi level and sharply reduce the Schottky-barrier height, Φ_B_, at MoS_2_/graphene contacts [34]. A similar vacancy-induced band-bending has been exploited in CZTS photovoltaics, where nanoscale MoS_2_ layers act as back-surface fields: Banerjee et al. demonstrated via comprehensive device modelling that controlled V_s_ densities of 10^12^–10^13^ cm^−2^ lower interface recombination and shift Φ_B_ by more than 0.2 eV [35]. Abo El-Ezz et al. further confirmed, by XPS and simulation, that V_s_ accumulation at MoS_2_/CZTS hetero-interfaces tunes carrier selectivity [36]. Extrapolating these findings to our lateral geometry, the forward bias of +6 V drifts pre-existing V_s_ toward the MoS_2_/graphene junction, narrowing Φ_B_ and switching the device from a high-resistance to a low-resistance state. The ON/OFF ≈ 2 is thus fully consistent with a barrier-controlled, rather than filamentary, mechanism [37].

Atomic-resolution STEM + EELS has visualized exactly this process: Hus et al. tracked an Au atom drifting into and out of a single S-vacancy in monolayer MoS_2_, producing reversible ON/OFF ≈ 3 switching via local Φ_B_ modulation [38]. Conductive-AFM noise spectroscopy further shows that individual V_s_ sites hop between neutral and negatively charged states on sub-second timescales, directly linking vacancy charge to lateral-device hysteresis [39].

## 4. Discussion

The present study demonstrates that a transfer-free, magnetron sputtering + confined-sulfurization route can convert sub-nanometer amorphous MoS_2_ precursors into continuous, few-layer 2H-MoS_2_ directly on CVD graphene without introducing detectable contamination or additional carbon-lattice disorder. Raman analysis (Δω ≈ 21 cm^−1^) and AFM roughness (S_q_ ≈ 0.8–0.9 nm) place the films squarely in the three-to-four-layer regime, consistent with seminal thickness calibrations by Lee et al. (Δω = 20–22 cm^−1^ for trilayer MoS_2_) [1]. The inability of vertical-geometry GIXRD to resolve MoS_2_ peaks is therefore attributed to the intrinsic detection limit for ≤2 nm layers rather than to incomplete crystallization. The appearance of resonant LA(M)–TA(M) (≈178 cm^−1^) and 2LA(M) (≈456 cm^−1^) Raman bands (Figure 3b) is a hallmark of long-range order in 2H-MoS_2_ and is absent in amorphous Mo–S [28,29,40]. Second, the Mo 3d_5_/_2_ full width at half maximum (FWHM = 1.03 eV) matches values reported for crystalline few-layer MoS_2_ and is much narrower than the 1.4–1.6 eV typical of amorphous phases [41]. These signatures confirm that both the 0.8 nm and 1.0 nm films are fully crystallized, even though their X-ray scattering volume lies below the GIXRD detection threshold.

Electrically, the lateral heterojunction yields a low-bias conduction hierarchy, I_graphene_ > I_graphene–MoS2_ > I_MoS2_. The measured graphene sheet resistance (~1.9 kΩ sq^−^^1^ after the 800 °C dwell) lies within the typical few-kΩ sq^−^^1^ range reported for large-area monolayer/few-layer CVD graphene after post-treatments, and is consistent with literature values for wafer-scale films and directly grown graphene on SiO_2_ [4]. Under ±7 V sweeps the heterostructure switches non-volatility with an ON/OFF ≈ 2.1, comparable to trap-assisted, barrier-modulated MoS_2_ memristors in the sub-5 nm thickness regime (ON/OFF < 10, V_SET_ ≈ 5–6 V) described by Shen et al. [16] and Xiong et al. [17]. In these devices, vacancy drift or charge trapping at the MoS_2_/metal (or MoS_2_/graphene) interface narrows the Schottky barrier, producing modest resistance windows yet excellent analog tunability, attributes desirable for neuromorphic weighting [33]. The identical mechanism is corroborated here by the small hysteresis loop at ±3 V, the SET at +6 V, and the partial RESET upon polarity reversal, all captured schematically in Figure 7.

From a fabrication perspective, the confined-space graphite box enables efficient sulfur usage (<30 mg) and eliminates toxic H_2_S, while the sputtered precursor affords Å-level thickness control. Compared with open-tube CVD of MoS_2_ on transferred graphene, which often yields uncontrolled multilayer islands or polymer-contaminated interfaces, the present route forms a chemically clean, laterally uniform heterostructure in a single furnace cycle. Li et al. demonstrated monolayer MoS_2_ growth in a sealed “sandwich” reactor [10]; our results extend the confined concept to wafer-compatible sputtering, producing smoother films (S_q_ ≈ 0.8 nm) than the wrinkled 20 nm RMS MoS_2_ obtained in similar two-step sputter systems [16].

Although the ON/OFF ratio extracted from Figure 6b is modest (≈2.1), such values are typical of vacancy- or barrier-modulated lateral MoS_2_ devices designed for analogue neuromorphic weighting rather than binary storage. Table 2 benchmarks our result against recent peers that employ sub-5 nm MoS_2_ channels and rely on interface charge trapping or S-vacancy drift rather than conductive filaments. Shen et al. reported ON/OFF ≈ 2.0 at V_SET_ ≈ 6 V in single-crystal monolayer MoS_2_ synapses and explicitly attributed the switching to reversible Schottky-barrier modulation [16]. Xiong et al. achieved ON/OFF ≈ 2–3 in oxygen-passivated few-layer MoS_2_, again through controlled interface-trap filling rather than filament formation [17]. These results confirm that our device performance lies within the accepted range for analogue 2D memristors. Comparable vacancy-drift lateral MoS_2_ memristors achieve 10^3^–10^4^ endurance cycles and >10^3^ s retention once lithographic contacts and encapsulation are implemented [16,17].

Moreover, modest resistance windows can be advantageous for multi-level conductance updates because they afford fine, symmetric weight tuning without abrupt jumps that plague high-contrast filamentary cells. Future optimization could explore intentional S-vacancy engineering (e.g., gentle Ar-plasma treatment) or dual-gate electrostatic modulation to enlarge the resistance window, or the insertion of an ultrathin h-BN tunnelling layer to suppress leakage or lower-temperature plasma-assisted sulfurization (≤450 °C) to render the process fully back-end-of-line compatible [42,43]. Additionally, integrating optical gating, as demonstrated for MoS_2_/graphene photomemristors [17], could yield multifunctional nodes that combine sensing and weight storage in a single laterally patterned element.

From a semiconductor-manufacturing perspective, two integration aspects merit comment. First, although the present demonstration used an 800 °C dwell to maximize crystallinity, plasma-assisted or rapid thermal sulfurization has recently achieved fully crystalline 2H-MoS_2_ at ≤450 °C on 300 mm wafers, within the thermal budget of advanced Cu/low-κ back-end stacks [44]. Because the amorphous precursor is deposited at room temperature, the peak process temperature can therefore be reduced without altering film-thickness control. Second, the graphite box confines the sulfur source (<30 mg) and suppresses ambient sulfur partial pressure, mitigating Cu embrittlement. Established diffusion barriers such as few-layer h-BN or sputtered TaN block S ingress below the 10^13^ atoms cm^−2^ are limited at 400 °C [14], and they can be inserted with no change to the two-step flow.

Overall, the results position the magnetron sputtering + confined-sulfurization approach as a simple, CMOS-scalable path to lateral MoS_2_/graphene memories, bridging the gap between fundamental 2D heterostructure physics and manufacturable neuromorphic hardware.

## 5. Conclusions

In this work, we have demonstrated that a simple two-step route, Å-level RF sputtering of an amorphous MoS_2_ precursor, followed by confined-space sulfurization at 800 °C, can fabricate centimeter-scale MoS_2_/graphene lateral heterostructures without resorting to polymer-assisted layer transfer or toxic H_2_S. Raman spectroscopy and AFM show that the converted film is uniformly three to four S–Mo–S trilayers thick, while XPS confirms an almost-ideal Mo:S ratio of 1:2 with only a thin surface oxide. The underlying graphene preserves its pre-anneal defect density and attains a sheet resistance of about 1.9 kΩ sq^−1^. Electrical measurements reveal a reproducible, non-volatile SET transition at approximately +6 V; the read-state current increases by a factor of two, a behavior that we attribute to sulfur-vacancy accumulation and attendant narrowing of the MoS_2_/graphene Schottky barrier. Although the ON/OFF ratio is modest, it matches values reported for other sub-5 nm MoS_2_ devices in which barrier modulation, not filament formation, dominates the resistive change, and is already sufficient for analogue synaptic weighting in neuromorphic circuits. The structural, chemical and electrical data position the magnetron sputtering + confined-sulfurization protocol as an industry-compatible route to wafer-level 2D memristive elements. The minimal precursor load, the elimination of wet-transfer polymer cleaning, and the use of a single standard furnace step together make the methodology economically attractive for wafer-scale integration.

## Figures and Tables

**Figure 1 nanomaterials-15-01239-f001:**
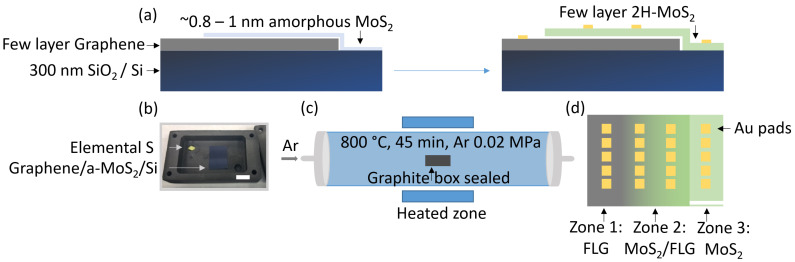
Two-step fabrication route of centimeter-scale MoS_2_/graphene heterostructures. (**a**) Cross-sectional schematic of the stack: few-layer graphene (FLG) on Si/SiO_2_ is coated with a ∼0.8–1 nm sputtered amorphous MoS_2_ film that is subsequently crystallized into few-layer (2–4 L) 2H-MoS_2_ during confined-space sulfurization; Au pads are added later for probing. (**b**) Photograph of the sealed graphite box loaded with elemental sulfur and the MoS_2_-coated FLG/Si coupon before annealing (1 cm scale bar). (**c**) Quartz-tube furnace used for sulfurization at 800 °C for 45 min in flowing Ar/S vapor; the graphite box sits in the uniform-temperature hot zone. (**d**) Top-view layout of the 1.5 cm × 2 cm coupon divided into a graphene-only region, a MoS_2_/graphene overlap region, and a MoS_2_-only region, each patterned with Au contacts (5 mm scale bar).

**Figure 2 nanomaterials-15-01239-f002:**
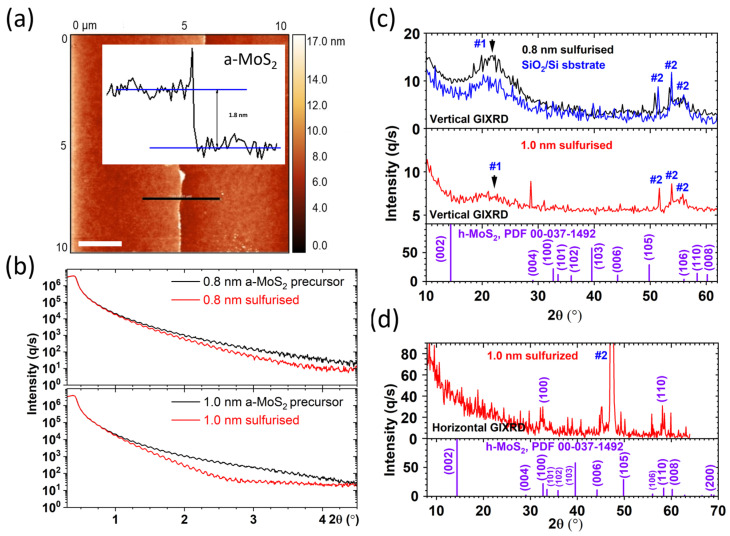
Thickness calibration and crystallographic evolution of sputtered amorphous MoS_2_ films during confined sulfurization. (**a**) Atomic force-microscopy maps and corresponding step-height profiles for 67 s deposition runs give a linear growth rate of 0.027 nm s^−1^, enabling precursor thicknesses of 0.8 nm and 1.0 nm (2 micrometer scale bar). (**b**) XRR curves for each precursor (grey) and its sulfurized counterpart (red): the appearance of pronounced Kiessig fringes after annealing confirms that the amorphous layers densify into continuous, few-nanometer 2H-MoS_2_ films. (**c**) Grazing-incidence X-ray diffraction collected in the vertical geometry (incident angle: 0.20°). For both the 0.8 nm and 1.0 nm sulfurized samples, the diffractogram is dominated by the broad amorphous SiO_2_ band and the Si (100) substrate peak (#1). Additional substrate peaks can also be observed (#2); no distinct MoS_2_ reflections can be resolved, and this result is attributed to the sub-2 nm film thickness being below the detection limit of the configuration. (**d**) Horizontal-beam GIXRD further reveals strong ⟨00l⟩ texture in the 1.0 nm film: only the in-plane (100) and (110) reflections are detected, indicating MoS_2_ sheets lying parallel to the substrate.

**Figure 3 nanomaterials-15-01239-f003:**
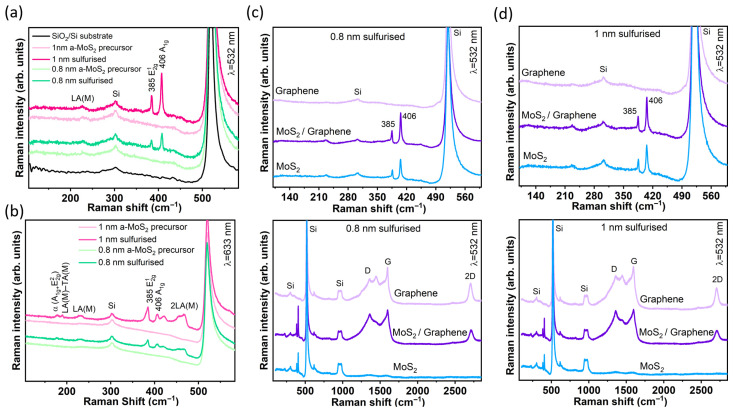
Raman evidence for crystalline few-layer MoS_2_ and formation of the MoS_2_/graphene heterostructure. (**a**) The 532 nm Raman spectra of the 0.8 nm and 1.0 nm sputtered amorphous MoS_2_ films before and after sulfurization. The crystallized films exhibit the E^1^_2g_ (~385 cm^−1^) and A_1g_ (~406 cm^−1^) modes of 2H-MoS_2_, separated by Δω ≈ 21 cm^−1^, consistent with 2–4 layers; the amorphous precursors show no MoS_2_ signatures. (**b**) Under resonant 633 nm excitation, the sulfurized films display strong second-order bands, LA(M)+TA(M) (~227 cm^−1^) and 2LA(M) (~460 cm^−1^), confirming good crystallinity. (**c**) The 532 nm point spectra on the 0.8 nm coupon demonstrate that graphene G/2D peaks appear only in the graphene-only and overlap zones, while MoS_2_ E^1^_2g_/A_1g_ modes dominate in the MoS_2_-only and overlap zones, verifying spatially selective heterostructure formation. (**d**) Equivalent three-zone spectra for the 1.0 nm sample show the same behavior with identical Δω, indicating a comparable few-layer thickness. Together, Raman mapping confirms conversion of ultrathin amorphous MoS_2_ into well-ordered 2H-MoS_2_ and preservation of an intimate, chemically clean MoS_2_/graphene interface.

**Figure 4 nanomaterials-15-01239-f004:**
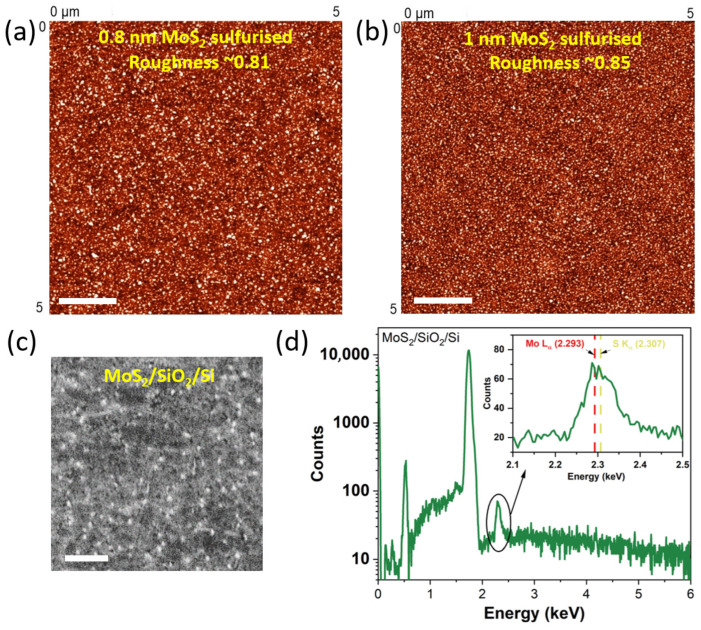
Surface morphology and stoichiometry of the sulfurized MoS_2_ films and the MoS_2_/graphene heterostructure: (**a**,**b**) 5 µm × 5 µm non-contact AFM height maps of the 0.8 nm and 1.0 nm precursor films after confined sulfurization. Both samples form continuous layers with areal root-mean-square roughness S_q_ ≈ 0.81 nm and S_q_ ≈ 0.85 nm, respectively (1 micrometer scalebar). (**c**) High-resolution SEM (20 k×) of the MoS_2_/FLG overlap region (0.8 nm sample) displays a compact nanograined surface composed of 30–50 nm crystallites; no cracks or pin-holes are observed, confirming full areal coverage of the underlying graphene (1 micrometer scalebar). (**d**) Energy-dispersive X-ray spectrum acquired from the same heterostructure (0–6 keV, logarithmic scale). The dominant Mo Lα and S Kα peaks overlap at 2.293–2.307 keV; deconvolution yields a sulfur-deficient atomic ratio due to the limited interaction volume and peak superposition in EDS. No extraneous elements are detected above background, corroborating the chemical purity of the confined-sulfurization process.

**Figure 5 nanomaterials-15-01239-f005:**
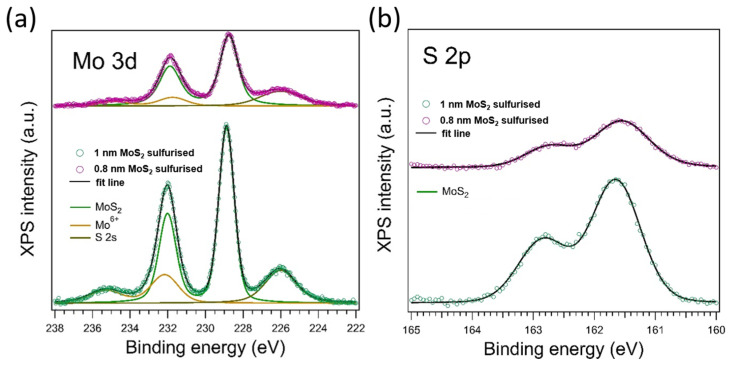
Chemical-state and stoichiometric analysis of confined-sulfurization MoS_2_ films. (**a**) High-resolution Mo 3d + S 2s spectra for the 1.0 nm (green) and 0.8 nm (magenta) films after sulfurization. Both are dominated by the Mo^4+^ doublet of 2H-MoS_2_ at 228.5/231.6 eV; a minor Mo^6+^ shoulder (yellow fit component) reflects surface oxidation acquired during air exposure. (**b**) Corresponding S 2p spectra show the S^2−^ 2p_3_/_2_/2p_1_/_2_ doublet at 161.6–162.8 eV with negligible high-binding-energy sulphate contribution, confirming complete sulfurization without Mo–O–S bonding.

**Figure 6 nanomaterials-15-01239-f006:**
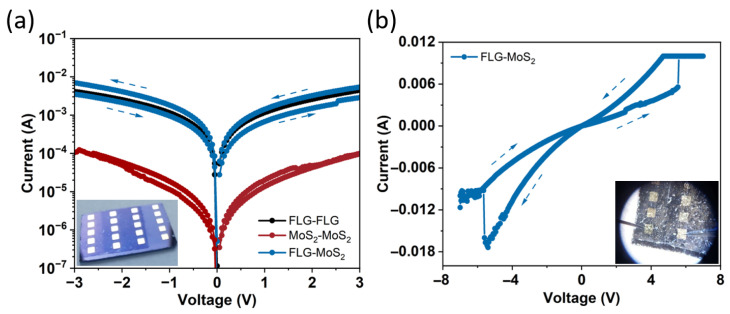
Electrical transport and memristive switching in the directly grown MoS_2_/graphene heterostructure. (**a**) Semi-log |I|–V curves recorded between −3 V and +3 V for three in-plane contact configurations. Graphene–graphene (FLG–FLG, black) is nearly ohmic; MoS_2_–MoS_2_ (red) remains four to five orders of magnitude less conductive within this bias window; the mixed graphene–MoS_2_ pair (blue) is intermediate and exhibits a small volatile hysteresis, consistent with reversible Schottky-barrier modulation at the MoS_2_/graphene junction. The inset shows the Au-pad array used for all measurements. (**b**) Extending the graphene–MoS_2_ sweep to ±7 V reveals a non-volatile SET event at ≈+6 V and a partial RESET on the negative branch. The resulting low-bias read current (0.5 V) changes by a factor ≈2.1, confirming true memristive switching localized in the crystallized MoS_2_ layer. Inset: Microscope image of the probed contact pair. All measurements were performed at room temperature in ambient air with a 10 mA compliance current.

**Figure 7 nanomaterials-15-01239-f007:**
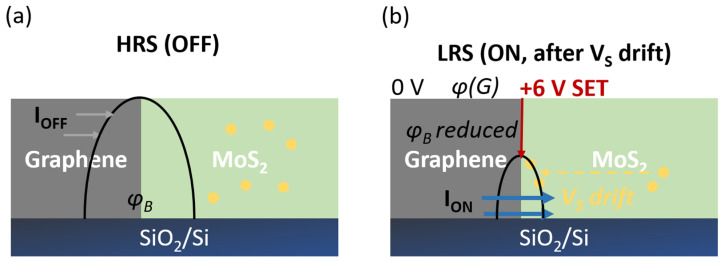
Conceptual model of vacancy-mediated switching in the lateral MoS_2_/graphene heterostructure. (**a**) HRS (OFF): Randomly distributed sulfur vacancies (yellow dots) reside in the MoS_2_ bulk; the MoS_2_/graphene interface is separated by a wide Schottky barrier, Φ_B_, so only a small leakage current, I_OFF_, flows. (**b**) LRS (ON): A positive bias of +6 V drives V_s_ toward the interface (dashed yellow arrow), forming a narrow defect-rich region that electrostatically pins the graphene Fermi level and reduces Φ_B_. The lowered barrier permits a higher tunnelling current, I_ON_. Removal or reversal of the bias allows for partial vacancy back-diffusion, thereby restoring a higher barrier and resetting the device.

**Table 1 nanomaterials-15-01239-t001:** High-resolution XPS fitting parameters for 0.8 nm and 1.0 nm MoS_2_ films. The 1 nm film is close to stoichiometric (S:Mo ≈ 1.95; Mo^6+^ ≈ 22%), while the thinner 0.8 nm film shows a slight sulfur deficiency (S:Mo ≈ 1.5; Mo^6+^ ≈ 14%). Overall, the confined-space process yields nearly stoichiometric 2H-MoS_2_ with only a thin native oxide layer and no detectable contaminants.

	1 nm MoS_2_ Sulfurized	0.8 nm MoS_2_ Sulfurized
S:Mo	1.95	1.5
Mo^6+^ (%)	22	14
B.E. Mo 3d (eV)	229	228.8
B.E. S 2p (eV)	161.7	161.6

**Table 2 nanomaterials-15-01239-t002:** Benchmark of lateral MoS_2_/graphene memristors.

Device Architecture	t_MoS2_ (nm)	V_SET_/V_RESET_ (V)	ON/OFF Ratio	Mechanism	Ref.
Ti/Au/monolayer MoS_2_/Ti/Au	≈0.7	+6/−5	≈2.0	Schottky-barrier narrowing by S-vacancy redistribution	[16]
TiN/O-doped 3 L MoS_2_/TiN	≈1.8	+5.4/−4.8	≈2–3	Oxygen-induced trap filling/interface dipole	[17]
Au/4 L MoS_2_/graphene/Au	≈2	+6/−6	≈2.1	Interface vacancy drift/barrier modulation	this work

## Data Availability

The original contributions presented in this study are included in the article. Further inquiries can be directed toward the corresponding author.

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
