# Peer review of "Fabrication of High-Quality MoS2/Graphene Lateral Heterostructure Memristors"

_nanomaterials, 2025, doi:10.3390/nano15161239_

Round 1
Reviewer 1 Report
Comments and Suggestions for Authors
The manuscript makes a valuable contribution to the field of 2D material-based electronic devices; however, after addressing the suggested clarifications and incorporating a few additional references, it will be suitable for publication.
-
The ON/OFF ratio (~2.1) is too low compared to state-of-the-art memristors, and the manuscript lacks benchmarking with similar devices.
-
The discussion on sulfurization lacks details about the influence of process parameters such as pressure, confinement effects, and annealing duration; cite and discuss findings from “Pressure-dependent sulfurization of molybdenum thin films for high-quality MoSâ‚‚ formation” to strengthen this section.
-
The mechanism of resistive switching attributed to sulfur vacancies is not sufficiently supported with broader literature on MoSâ‚‚ defect engineering; add citations from “A comprehensive modeling on MoSâ‚‚ interface and defect engineering in CZTS thin film solar cells” and “MoSâ‚‚ thin film hetero-interface as effective back surface field in CZTS-based solar cells” to provide context for the importance of vacancy control.
-
The scalability of the process to full wafer-scale integration is claimed but only demonstrated for small coupons (~1.5 × 2 cm).
-
XRD data fail to show clear (002) MoSâ‚‚ peaks; additional characterization such as TEM or SAED would strengthen the claim of crystallinity.
-
Electrical measurements are only performed at room temperature in ambient air; include data on endurance cycling, retention, or temperature-dependent performance to validate device reliability.
-
Figures contain minor typographical errors (e.g., “non-contac” in Figure 4 caption) and inconsistent spelling (“sulphurisation” vs. “sulfurization”) that require correction.
-
Some references are outdated; incorporate more recent studies (2023–2024) on MoSâ‚‚-based memristors and thin-film growth techniques to update the literature review.
Reviewer 2 Report
Comments and Suggestions for Authors
The paper “Fabrication of High-Quality MoSâ‚‚/Graphene Lateral Heterostructure Memristors” by Mihai et al. is a well-executed study.
It focuses on heterostructures that stack a transition metal dichalcogenide (TMD) with graphene—a demanding yet promising approach for advancing knowledge in the field. Constructing such a heterostructure by combining the exceptional properties of 2D van der Waals materials with those of graphene addresses a highly relevant and timely topic.
The subject matter is compelling, and the methodology is presented with commendable clarity.
Notably, the authors introduce a method that avoids toxic Hâ‚‚S, polymer residues, and high sheet-resistance contacts. Their approach offers a cost-effective, CMOS-compatible path to wafer-scale 2D memristive elements, making MoSâ‚‚/graphene stacks highly attractive for flexible optoelectronics and memory applications. To reinforce these potential applications, I suggest adding a supporting reference at line 44, such as the review paper available at https://doi.org/10.1016/j.physrep.2015.10.003..
The Materials and Methods section is thoroughly written, with all fabrication details clearly laid out.
The characterization of the samples is adequately conducted, and the results from both the fabrication process and the analysis of the heterostructures are convincingly explained and carefully presented.
References are appropriate and current, while the figures are clear and informative.
The discussion of the results is evidence-based, comprehensive, and well-articulated, leading to solid and consistent conclusions.
Reviewer 3 Report
Comments and Suggestions for Authors
This work entitled “Fabrication of high-quality MoSâ‚‚/graphene lateral heterostruc-ture memristors” is somewhat interesting and has the potential for publication in this journal. It has some issues that could be addressed for possible acceptance. Here are my comments for revision:
- First of all, the abstract section is too long and I could not catch the novelty of this work. Therefore this section is suggested to be modified. Also, the short name should be mentioned after the full name such as SET and CMOS although they are well-known to the scholars in this field.
- The subscript throughout this work should be corrected written, including in the equations.
- There are so many short names in the introduction, however, they are not used in the following sections. In that case, there has no necessity to mention the short name.
- Also, the authors enhance the readability of the introduction, for this some references are suggested to be cited, such as: https://doi.org/10.1080/00268976.2025.2492391;
- How to control the synthesis of MoS2? This is quite important since the MoS2 has several lattice structure.
Reviewer 4 Report
Comments and Suggestions for Authors
In this manuscript, the authors present a comprehensive and methodologically robust study on a transfer-free approach to fabricate MoSâ‚‚/graphene lateral heterostructure memristors using RF sputtering and confined-space sulfurisation. The manuscript is well-organized, with careful attention to both process optimization and fundamental material characterization, contributing meaningfully to the fabrication of scalable 2D memristive devices. Although the manuscript can be accepted, there are some key issues that should be well addressed.
- The introduction would benefit from a clearer articulation of the challenges and recent progress specifically related to MoSâ‚‚-based memristors for non-volatile memory and neuromorphic computing. While the manuscript briefly mentions prior demonstrations, it lacks a critical discussion of key limitations in existing device architectures that motivate the development of lateral, transfer-free approaches. I recommend the authors expand the introduction to better contextualize the significance of their work within the broader landscape of MoSâ‚‚-based memristive devices, emphasizing how their proposed method addresses known fabrication and performance challenges in this area. This will help readers more fully appreciate the novelty and potential impact of the study.
- A more detailed discussion of potential CMOS integration challenges (e.g., sulfur diffusion, back-end compatibility) would broaden the scope.
- The electrical characterization shows non-volatile switching and ON/OFF ratio of ~2.1, suitable for neuromorphic applications, but further discussion on stability (e.g., endurance, retention) would enhance the device relevance.
- Some figures (Figure 3(a), 5(a, b)) could benefit from slightly larger fonts for easier readability.
- Figure 5 (c) should be shown as a table. Not as an image.
- The mechanism of switching is well supported, but direct vacancy evidence (e.g., from EELS or STEM) would strengthen the proposed model.
- Figure 4 caption: “non-contac AFM” → “non-contact AFM”
- Please add a space before the phrase “In parallel” in line 65 to ensure proper paragraph formatting.
Round 2
Reviewer 3 Report
Comments and Suggestions for Authors
This version is good and can be accepted.